

# Construction of a user-friendly software-defined networking management using a graph-based abstraction layer

Yufeng Jia[1,2], Jiadong Ren[2], Xianshan Li[2], Haitao He[2], Pengwei Zhang[1,2] and Rong Li[1]

[1] School of Information Science and Engineering, Xinjiang College of Science and Technology, Korla, Xinjiang, China
[2] School of Information Science and Engineering, Yanshan University, Qinhuangdao, Hebei, China

## ABSTRACT

The software-defined networking (SDN) paradigm relies on the decoupling of the control plane and data plane. Northbound interfaces enable the implementation of network services through logical centralised control. Suitable northbound interfaces and application-oriented abstractions are the core of the SDN ecosystem. This article presents an architecture to represent the network as a graph. The purpose of this architecture is to implement an abstraction of the SDN controller at the application plane. We abstract all network elements using a graph model, with the attributes of the elements as the attributes of the graph. This virtualized logical abstraction layer, which is not limited by the physical network, enables network administrators to schedule network resources directly in a global view. The feasibility of the presented graph abstraction was verified through experiments in topological display, dynamic route, access control, and data persistence. The performance of the shortest path in the graph-based abstraction layer and graph database proves the necessity of the graph abstraction layer. Empirical evidence demonstrates that the graph-based abstraction layer can facilitate network slicing, maintain a dependable depiction of the real network, streamline network administration and network application development, and provide a sophisticated abstraction that is easily understandable to network administrators.

# INTRODUCTION

Software-defined networking (SDN) serves as an innovative network architecture that decouples the data-forwarding logic from the control logic (*Ahmad & Mir, 2021*). The management and decision-making logic for the whole network are transferred to the control layer (the Controller). The data forwarding layer takes action based on the control layer's decision-making and is responsible for general data forwarding (the SDN architecture is shown in Fig. 1) (*Yan et al., 2021*). This decoupling of the control plane from the data forwarding layer resolves the tight coupling problem between the two planes in traditional distributed network architectures. This "programmable network" approach enables network functions to be implemented as controller applications on the application

Corresponding author
Jiadong Ren,
jiayufeng@stumail.ysu.edu.cn

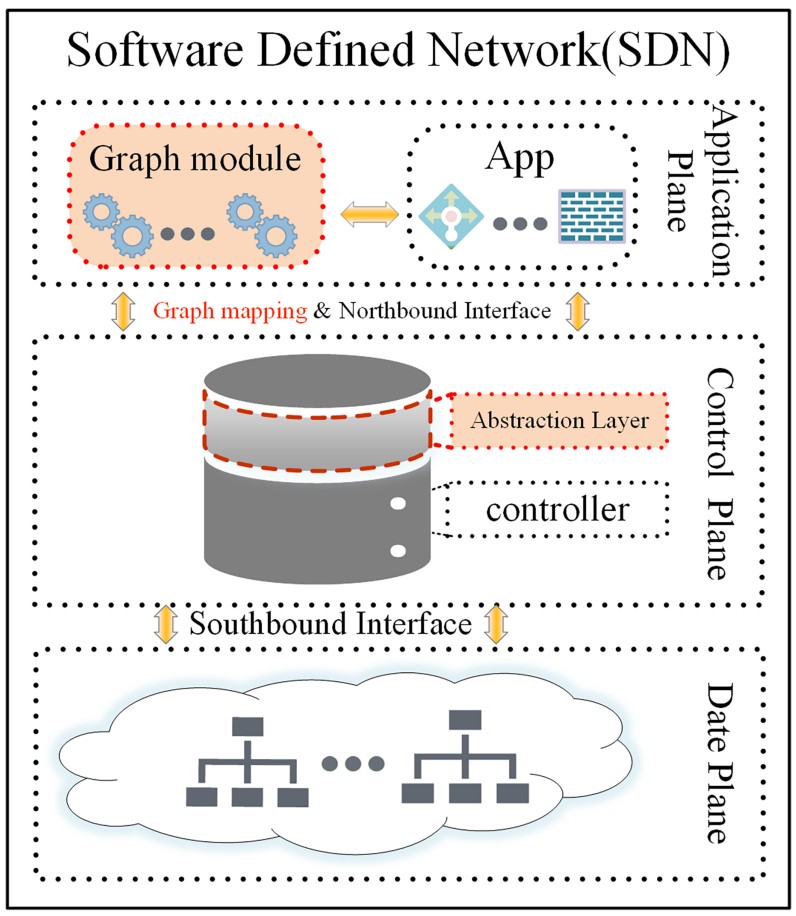

**Figure 1 The SDN architecture, abstraction layer, and graph model.**

plane. These applications enable the SDN controller to respond quickly to changes in the data plane. The SDN paradigm, with its convenient management, scalability, flexibility, and monitoring capabilities, has revolutionized the way networks are implemented and managed, becoming one of the most important network architectures with a wide range of applications in data centres, cloud computing, Internet of Things, security, and other areas (*Bhuiyan et al., 2023*). SDN employs application programming interfaces (API) to facilitate communication between planes (*Latif et al., 2020*). The southbound interface abstracts hardware complexity, enabling seamless communication between the control and data planes. The main protocols include OpenFlow (*Open Networking Foundation, 2024*), NETCONF, OVSDB, and others. Among these, OpenFlow protocol has become the defacto international industry standard (*Xu, Wang & Xu, 2019*). Unlike the southbound interface, the northbound interface (which enables interaction between the control plane and the application plane) still lacks an industry-recognized standard (*Alghamdi, Paul & Sadgrove, 2021*; *Jiménez et al., 2021*). Additionally, the east-west interface used for communication between controllers has not received sufficient attention from the industry (*Cox et al., 2017*; *Bannour, Dumbrava & Lu, 2022b*). This application interface abstraction

provides a helpful way to avoid writing network programs and policies at a lower level. This article focuses primarily on the abstraction provided by the northbound interface.

In response to the challenges of northbound abstraction, Pyretic (*Reich et al., 2013*) introduces simple, reusable high-level abstractions and proposes functional modularization as an effective approach for constructing SDN control applications. Ryuretic (*Cox et al., 2016*) provides an additional layer of abstraction on top of the Ryu controller (*Nippon Telegraph and Telephone Corporation, 2014*) that couples multiple modules with existing applications to forward, redirect, discard, or create packets, allowing users to interact with packets actively and passively.

In recent years, Nepal (*Jamkhedkar et al., 2018*) has proposed a hierarchical network model based on a graph database. This model enables the representation and inference of network service topologies, as well as the reasoning of data flows within the network. Leveraging these capabilities, it supports automated network application management through a dynamically configurable, cloud-based virtual network. Similarly, SeaNet (*Souza et al., 2015*) introduces a knowledge graph-driven approach designed to facilitate autonomous network management in SDN environments.

Although there are various northbound abstractions with different purposes for targeting different network strategies, the requirements of network policies are constantly changing. As a result, network administrators need to combine the advantages of various strategies to achieve their goals (*Bhuiyan et al., 2023*; *Alghamdi, Paul & Sadgrove, 2021*; *Bannour, Dumbrava & Lu, 2022b*). Moreover, different abstractions use different data structures and definitions, and in order to combine multiple abstractions, these data need to be transformed, a transformation that becomes more and more complex as the amount increases. This dramatically increases the complexity of management and also requires higher levels of skill for network administrators. Consequently, this creates challenges for advancing the universality of SDN.

In the early days of the development of SDN, it was necessary to manage complexity. However, now it is necessary to extract simplicity to popularize and apply it better. Two major challenges remain:

- The primary function of the northern interface is to facilitate the connection between a lower-level interface and a higher-level junction. How can we propose an easy-to-write high-level abstraction (or data structure) to express the inner workings of the network while being able to cover a larger abstraction space?
- How can we make the advanced interface more efficient and in tune with the human mindset, while still able to keep up with the changing network policies?

The SDN paradigm relies on decoupling the control and data plane and logically centralizing control (*Ahmad & Mir, 2021*), thus enabling direct network programming *via* open interfaces, which are the core and key. To address these challenges, we found that the global view is the key advantage in solving the abstraction of the northbound interface through detailed thought and analysis. Gavel (*Barakat, Koll & Fu, 2019*) is an SDN

controller whose core is based on the graph database to represent the global network topology. It leverages the graph database's robust support for graph structures, delivering significant performance improvements. Gavel served as a source of inspiration for us, and we also found that the network is essentially a graph structure in which the applications in the network deal with various relationships within the graph structure. Additionally, queries and filtering in the network may be considered as a form of pruning graph. This article proposes using the graph model to build a dynamic abstraction layer for network representation and management, which significantly simplifies the development of applications and policies. The abstraction layer is shown in Fig. 1 (to be more accurate, we later call the abstraction layer a graph-based abstraction layer). Our concept was successfully validated through experiments on the open-source Ryu controller.

Our abstraction adheres to the fundamental principles of SDN without altering the current controller architecture; no additional equipment is needed, and it has excellent universality and superior performance. In summary, the primary contributions of this article are as follows:

- We envision that all the devices in the network have a logical counterpart, and we abstract them with graph models, allowing real-time interaction between the physical network and the virtual network. We also use the Neo4j graph database (Neo4j, Inc, 2024) for persistent interaction with the past states, and this logical abstraction can support network slicing, enriching our network applications significantly.
- Based on the graph abstraction and utilizing the various graph algorithms provided by the graph module, we verified different network applications to show how network administrators can easily program previously complex network management applications. We experimentally verified network topology visualization and shortest path forwarding based on delay quality, access control, data persistence, and other network management tasks. This novel abstraction pattern introduces new opportunities for network management and automation.
- Our experiments demonstrate that the graph abstraction model (graph-based abstraction layer) reliably represents the real network. It simplifies network management and application development without requiring detailed knowledge of the controller. It offers a high-level abstraction that matches human intuition.

The rest of the article is structured as follows: "Background and Related Work" introduces the northbound interfaces and graph abstract layers. "Design and Solution" includes construction principles and methods of network applications. "Results and Discussions" verifies the feasibility of the graph abstract layer. At last, "Conclusion" concludes our work and discusses the future.

## BACKGROUND AND RELATED WORK

The data plane is comprised of various physical infrastructures such as host machines, servers, switches, and routers interconnected with each other, which are mainly

responsible for data forwarding. The control plane manages data forwarding in the data plane and performs traffic control, statistical analysis, and other functions. Additionally, it provides a unified network service API for the application plane and the network abstraction to upper-layer applications, giving the network the capability of software programming. The application plane uses the northbound API to control and define the network, telling the network how to meet its business needs in a programmable way.

Suitable northbound interfaces and application-layer abstraction are the core of the SDN ecosystem and key to popularization and development at the social level. To this day, the suitable abstraction and neutral standards for various network manufacturers are still in the early stages (*Latif et al., 2020*; *Cox et al., 2017*).

## Northbound interface

The northbound interface of an SDN is an interface open to upper-layer applications *via* the controller, the goal of which is to enable applications to invoke the underlying network resources conveniently. Concerning the northbound interface of the SDN, the implemented functionality is identical to the connection between two software entities. That is, the northbound interface does not require any new protocols and communication between the application and control layers can be implemented by writing simple socket interface programming. Obviously, this is a lower-level, tedious, and time-consuming programming process that often requires complex refactoring by network administrators. Due to the lack of a standardized northbound interface, some controllers have proposed custom northbound interfaces specific to the controller (*Latif et al., 2020*; *Halder, Barik & Mazumdar, 2017*). Some controllers implement a northbound interface designed with a RESTful architectural style. For instance, controllers like Floodlight, ONOS, and Ryu utilize REST APIs to facilitate communication and management (*Xu, Wang & Xu, 2019*). However, developers still need to learn new APIs, data structures, and specific conventions of the controllers whenever they want to support a new controller or have new application requirements. Moreover, these interfaces may become verbose when used frequently and in large quantities. There are also specific requirements that need to be better supported (Ryu lacks REST API for obtaining host information).

In this article, we only aim to verify the feasibility of the graph-based abstraction layer and do not intend to replace the northbound interfaces of existing controllers. The northbound interface and the graph-based abstraction layer are in a parallel relationship, enriching the upper-layer applications invoke.

## Graph-based abstraction layer

The programming logic for communication between the control plane and the application plane is called the graph-based abstraction layer. The layer is designed directly to serve the needs of applications, so its design needs to be closely aligned with business requirements and have diverse features. How can network administrators control the resource state of the entire network in a software-programmed way and schedule it consistently? This abstraction requires a rethinking of the network architecture and management model.

Many abstract and strategy frameworks have been proposed to simplify network programming for different types of applications (*Halder, Barik & Mazumdar, 2017*) proposed a graph-based SDN flow conflict detection method. Any abstraction involves converting one representation to another. One challenge in creating effective abstractions is ensuring that they accommodate reasonable network distribution, provide convenient data representation, remain accessible for a wide range of network applications, and support core operations such as addition, deletion, modification, and querying. By addressing these challenges, abstractions enable users to express their goals at a higher level, thereby simplifying network management.

In the SeaNet (*Zhou, Gray & Mclaughlin, 2021*) system, recursive algorithms are used to read network records, extract semantic information, and store it in a knowledge base. Expert knowledge and network management rules can also be formalized as knowledge graphs. Through SPARQL automatic inference and network management API, SeaNet enables researchers to develop semantic intelligent applications on their own SDN. An autonomous network management system is implemented using the abstract approach of knowledge graphs. However, this abstract approach relies heavily on expert intervention during knowledge updates.

Ravel (*Wang et al., 2016*) implements the entire SDN network control infrastructure using a standard SQL database. The network abstraction is represented by user-defined SQL views, which are represented by SQL queries that can be dynamically added. Application developers can request views based on database tables for different applications. However, retrieving information that needs to be collected from many tables will lead to severe delays. Its advantage is at the cost of performance.

*Wu et al. (2020)* proposed a method to import Network Markup Language (NML) models into a scalable graph database (Neo4j) and use semantic modelling technology in graph databases to enhance the state of SDN networks. This enables SDN controllers to provide developer-friendly primitives, simplifying the design of SDN control applications. However, the most easily understandable representation of the network is in fact a structured graph.

GOX (*Bannour, Dumbrava & Danduran-Lembezat, 2022a*) is a proof-of-concept controller that implements the SDN architecture in a graph database. Gavel (*Barakat, Koll & Fu, 2017*, *2019*) is the first controller to use a graph database to generate data representations for an SDN. Gavel significantly reduces programming complexity, improves efficiency, and scales well across large networks. However, they are controllers and do not use graph models to achieve compatibility and joint management with existing controllers.

In summary, to make the existing controller and applications more compatible, efficient, and scalable, we propose a novel abstraction method, that places an additional abstract layer on the controller framework. The abstraction layer builds a real time network in the form of a graph and then interacts with applications in real-time in the form of graph mapping. Our goal is to simplify application development and create new opportunities for both management and automation.

## DESIGN AND SOLUTION

### Methodology

Modern network structures are dynamic, large-scale, cross-correlated, and constantly emerging with various needs and technologies. To manage and maintain such networks, it is necessary to understand network elements (such as servers, switches, virtual machines, and network functions) and their associative interrelationships (such as querying, analyzing, and modifying the flow of dynamic data between the different elements). So, how can such a complex network situation be abstracted in a straightforward, natural, and precise way? Common data structures such as arrays, stacks, linked lists, and tree structures are all exceptional types of graph structures. Graphs are high-dimensional, and the higher dimensions are downwardly compatible and represent lower dimensions, while the reverse is not as easy. That means, expressing high-dimensional relationships using low-dimensional relational data structures is a quite significant challenge. Analysis has found that networks are naturally graphical structures, and their abstraction as graphs does not require the re-implementation of complex data structures and algorithms, it is very visual. It is intuitive to access the data of nodes in the network and also avoids creating new dependencies between different structures. Therefore, abstracting the network as a graph and using existing graph modules (graph algorithms) and graph databases to interact with network applications, allows this graph abstraction to be used as the basic interface from the control plane to the application. Naturally, all types of network management will be carried out efficiently and easily.

### System architecture

We already know the system architecture of SDN from the previous introduction. How do we implement the idea of the "Methodology" section based on the SDN architecture by taking advantage of the features of SDN without making any physical changes? We present how to model physical and virtual resources in a graph $G = (N, E)$ where $N$(Nodes) is a set of physical and virtual resources and $E$(Edges) is their relationship. Edges and nodes provide different types of resources and services. We can define graph for nodes $G(N)$ as follows:

$$G(N) = G_p(n) \bigcup G_v(m) \quad N \geq n \tag{1}$$

where $n$ is the number of physical nodes, $m$ is the number of virtual nodes, $p$ is a set of physical resource mappings, and $\forall$ is a set of virtual resources. $G_p(n)$ is a graph for representing physical nodes and physical networks mappings for the given node $n$, whereas $G_v(m)$ is a graph for representing virtual nodes and virtual networks. So, $G(N)$ is a union of both graphs $G_p(n)$ and $G_v(m)$.

On the structure of the physical infrastructure (physical servers, physical switches, physical routers, and other physical devices) comprise, we introduce a virtualization logical abstraction layer with a one-to-one mapping based on the network topology (graph-based abstraction layer is $G(N) = G_p(n) \bigcup G_v(m) \quad N = n$) using the graph model. The operation of physical infrastructure enables the logical network to be instantiated in the

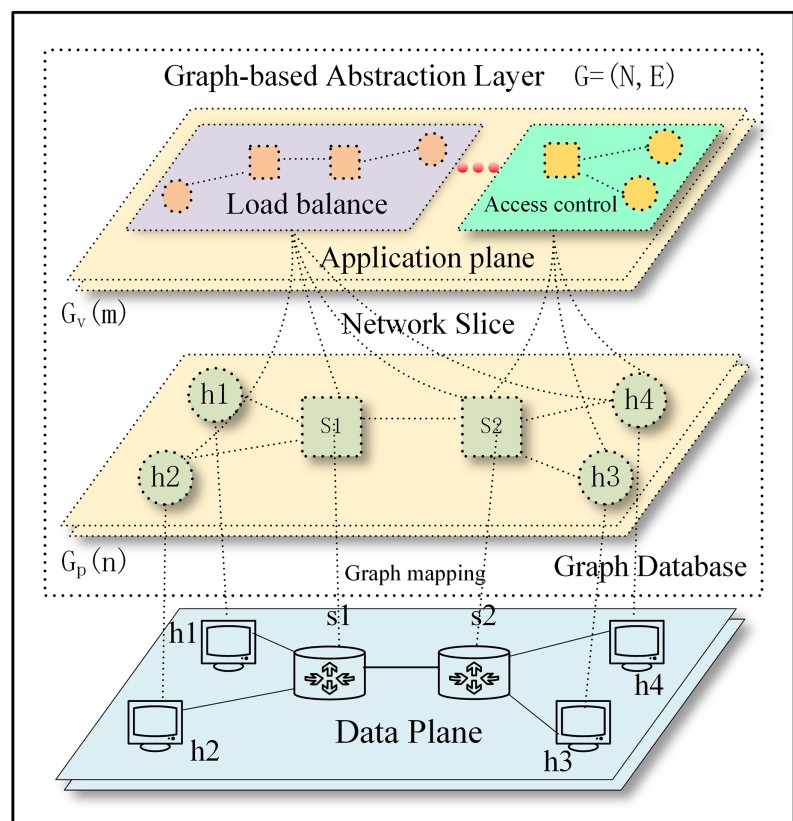

**Figure 2 The graph-based abstraction layer.**

form of a graph. The instantiated graph network can decouple from the physical network, thereby allowing for the virtualization and arbitrary combination of multiple logical networks. This enables network slicing to take place (graph-based abstraction layer is $G(N) = G_p(n) \bigcup G_v(m) \quad N \geq n, m \in \forall$).

The graph-based abstraction layer as shown in Fig. 2.

Applications operating at the application plane can seamlessly integrate with established third-party modules without the need to continually create fundamental functionalities. The graph-based abstraction layer is mapped with a graph structure, which can naturally call on existing graph modules (the graph module is shown in Fig. 1). The application plane satisfies the needs of different businesses and network innovations through querying, calculating, analyzing, and dynamically adjusting. Finally, it guides data forwarding in the physical plane based on business requirements.

## Implementation method

The following part will address the process of abstracting the physical network into a graph-based abstraction layer, preserving all network element information and network information flows. We treated the entire physical network as a graph, which is a set of nodes and edges. Next, we solved how to abstractly map the nodes, edges, and their attributes to the physical network, as shown in Eq. (1). The graph mapping of the

graph-based abstraction layer to the data plane and network slice is shown in Fig. 2. Considering scalability and generality, we saved attribute data as a dictionary so that users can easily add any necessary data. The graph-based abstraction layer is simplified to include the following components:

**Nodes:** also known as vertices. They contain network elements (servers, switches, network functions, and virtual machines) and are associated with a unique identifier of a network device already maintained by the controller.

**Attributes of nodes:** attributes associated with each node, each expressed as a key-value pair. Each node has multiple attribute fields. Attributes are optional and can be dynamically added, deleted, or modified. The name and category are considered special attributes.

To simplify the experimental model, we assumed that the attributes of a host included name, category, DPID number, PORT number, IP address, and MAC address. The attributes of a switch included name, category, source DPID number, destination DPID number, source PORT number, and destination PORT number. Figure 3 is a simple network model of our implementation method, and it includes node (switch and host), edge (edge and its direction), and attributes of node and edge.

Figure 3 was created using code to implement the demo of our graph-based abstraction layer, while also explaining what events and attributes are. The Ryu controller manages and controls network behavior by responding to events through defined event handler functions. When the controller detects a new switch connection in the network and the switch completes the handshake (*i.e.*, completes the initial OpenFlow protocol communication), the EventSwitchEnter event is triggered. The graph abstraction layer adds a switch node in this event function. Attributes of the switch node include name, dpid, ports, and links. Below, we demonstrate this with an example code, which is labeled as Code 2 (Eq. (2)) for reference:

$$
\begin{aligned}
&\texttt{SwitchGraph = nx.DiGraph()} \\
&\texttt{SwitcheGraph.add\_node(name, dpid = dpid, ports = ports, links = \{\})}
\end{aligned}
\tag{2}
$$

When updating switch attributes:

```
SwitcheGraph.nodes[name]['links'].update(value)
```

When we needed to add other nodes, such as adding a host. Here, mac, ip, and port are the attributes of the host, we simultaneously updated the attributes:

```
HostGraph = nx.DiGraph()
HostGraph.add_node(name, dpid = Hport.dpid, port = Hport.port_no,
ip = host.ipv4, mac = host.mac)
```

When querying host attributes:

```
Host = nx.get_node_attributes(HostGraph, 'dpid')
```

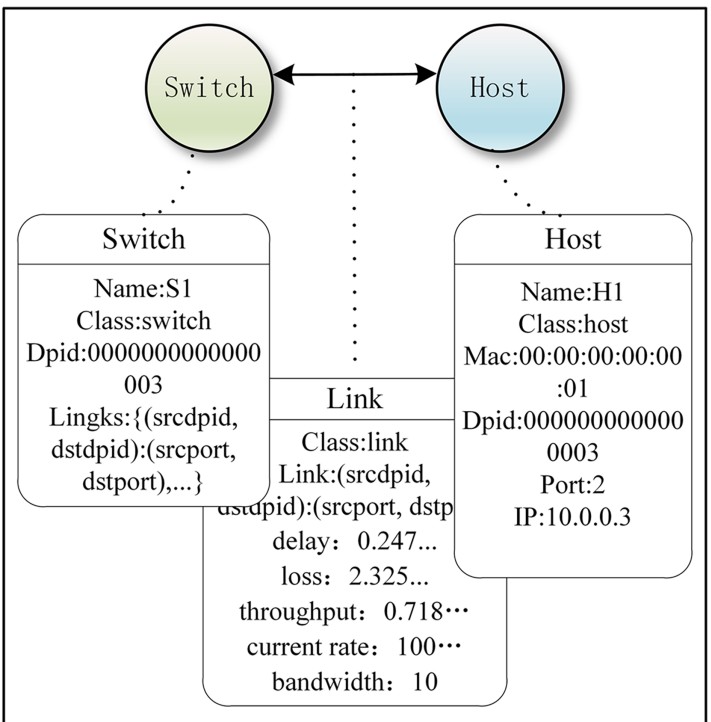

**Figure 3 Example of a simple network under our Implementation method.**

Add an edge between two switches, and update attributes such as latency, bandwidth, and throughput at the same time:

```
SwitchGraph.add_edge(src, dst, delay=link_delay,
loss=packet_loss, throughput=link_throughput,
curr_speed=CurrSpeed)
```

When you want to expand the graph-based abstraction layer, you can merge them together. This could be a merger of switch and switch, a merger of switch and host, or a merger of the graph-based abstraction layer and another graph-based abstraction layer, as well as the merger of the graph-based abstraction layer with network slicing. An example of the code implementation is shown below and is labeled as Code 3 (Eq. (3)).

$$
\text{Graph} = \text{nx.union(SwitcheGraph, HostGraph)} \tag{3}
$$

**Edges:** Usually called relationships, each edge connects a pair of nodes. There are two types of edges: one represents physical connections between network elements: "Link", and the other represents application connections between network slices and the graph-based abstraction layer: "Member".

**Attributes of edges:** Each attribute is expressed as a key-value pair. Multiple attribute fields can be stored. Attributes can be dynamically added, deleted, or modified. Name and category are considered special attributes. The attributes of an edge include delay, loss,

```
1
2   def get_links(self, Dpid=None):
3       link_lists = get_link(self.topology_api_app, Dpid)
4       for link in link_lists:
5           src = link.src
6           dst = link.dst
7           nodeattr=nx.get_node_attributes(self.SwitchGraph,'dpid')
8           for name,dpid in nodeattr.items():
9               if dpid==src.dpid:
10                  value={(src.dpid, dst.dpid):(src.port_no, dst.port_no)}
11                  self.SwitchGraph.nodes[name]['links'].update(value)
12
13          Source = self.dpip_map_name[src.dpid]
14          Destination = self.dpip_map_name[dst.dpid]
15
16          link_delay=self.get_link_delay(src.dpid,dst.dpid)
17          packet_loss=self.get_link_loss(src.dpid,dst.dpid)
18          CurrSpeed,MaxSpeed=self.get_port_speed(src.dpid,dst.dpid)
19          link_throughput=self.get_link_throughput(src.dpid,dst.dpid)
20
21          self.SwitchGraph.add_edge(Source,Destination,delay=link_delay,
22                                    loss=packet_loss,curr_speed=CurrSpeed,
23                                    throughput=link_throughput)
24
```

**Figure 4 The edges update attributes of delay, loss, throughput, bandwidth, current rate, and correspondence between DPID and PORT.**

throughput, bandwidth, and current rate. The connection between switches also includes the correspondence between DPID and PORT. Attributes are optional and can be dynamically added, deleted, or modified. The source code is shown in Fig. 4, and the attributes of the edges are updated.

**Direction of edges:** To simplify the experimental model, our edges have direction. For example: h1 → s1 and s1 → h1.

**Network slices:** In the graph model, the definition of network slices occurs after the graph abstraction object is created, and the objects contained in each slice can be dynamically adjusted. Distinct slicing patterns (application) can be defined for nodes and edges individually.

**Graph update:** The data of the graph, nodes, edges, and their attributes can be added, deleted, or modified dynamically. We updated the graph information (creation, modification, deletion, execution of algorithms, and data retrieval in the graph) by listening to the corresponding events. The source code in Fig. 4 shows how to add edge attributes.

**Flow Setup:** We modified the traditional flow setup to significantly reduce the traffic on the control channel. This improved flow setup was implemented in our subsequent experiments and is illustrated in Fig. 5 (with the path 0 → 1 → 2 → 3). When the ingress switch receives a packet, it attempts to match the packet against its flow table using the specified match field. If no match is found, the switch encapsulates the packet header in a "PacketIn" message and forwards it to the controller (*Xu, Wang & Xu, 2019*). The controller, leveraging the global view of the network topology, responds by generating multiple flow rules. These flow rules are then installed on all switches along the path using

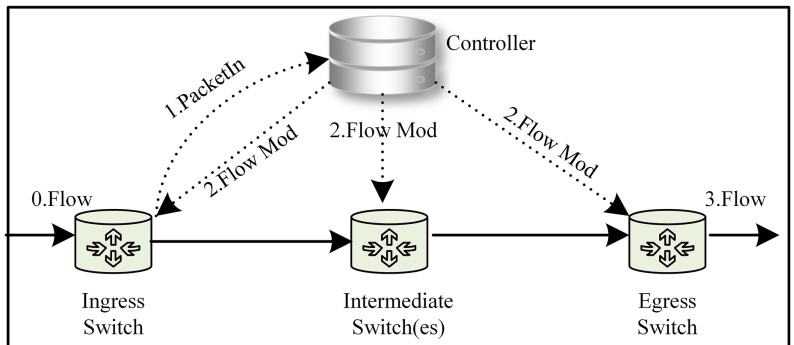

**Figure 5** An improved Flow Setup model was used in our experiment.

a "FlowMod" message. By installing the flow rules directly on the switches, the need for frequent communication between switches and the controller is reduced, thereby enhancing overall network performance.

## Network slice

The activation of the network slice occurs after the graph-based abstraction layer has been created. Because the graph abstraction is virtualized, various forms of logic can naturally be provided. This virtualized logic can be seen as the network slice. These virtual logics can meet different business requirements because the nodes in a virtual network can belong to a complex classification and belong to different applications. The network slice is shown in Fig. 2, from the high-level network functions in the abstract (such as Firewall) to the properties of nodes (such as PORT), we can dynamically adjust the nodes contained in each slice. Nodes and edges can be defined with their own application affiliations. It is, therefore, essential to classify and label these nodes and edges so that network operators and management systems can query and manage network elements at the appropriate level of abstraction.

The graph is a distributed structure that can be easily embedded in another graph or easily create a new node. The different classifications of nodes can be easily represented as instances of an application, and the application will be composed of nodes of the same type. The network application has a relationship "Member" with the node. This modular and hierarchical architecture allows network managers to customize the network and provide highly flexible virtual network services. Currently, in the application of network slice, except for the graph structure, other data structures cannot be so flexible.

## RESULTS AND DISCUSSIONS

As a proof-of-concept, the experiment aims to validate and evaluate the feasibility of the graph-based abstraction layer. We tested basic functions on networks with different topologies and scales, and we wrote network applications that followed graph abstraction. We will explain network topology structure, delay-based shortest path forwarding, data persistence, and access control:

## Experimental environment

Our experiment used an Intel(R) i7-7500U CPU operating at 2.70 GHz and a 16 GRAM PC running Ubuntu 20.04 LTS. In our experiment, Mininet 20 was used to build the underlying network, Ryu was used as the controller, NetworkX21 was used for the graph-based abstraction layer, and the Neo4j database was used for data persistence. The controller Ryu, simulation tool Mininet, and graph database Neo4j all have strong authenticity and can be applied in real-word scenarios without modifying the code.

To simulate the traffic of real networks, a more comprehensive evaluation of system efficiency and scalability was provided. In "Network Topology Verification", "Graph Module Verification", and "Data Persistence", we evaluated the system using randomly generated traffic of varying sizes from random hosts. In "Network Slice Verification", we evaluating with stable traffic generated by random hosts. Lastly, no traffic was involved in "Network Latency Discussion". Specifically, the size range of random traffic is as follows:

Mouse flows = ['100K', '200K', '300K', …, '9000K', '10000K', '1000K'].
Elephant flows = ['10M', '20M', '30M', …, '800M', '900M', '1000M'].

## Network topology verification

One of the characteristics of SDN is the ability to obtain a global network view. The global network view is important, as network administrators often need to use it to select and implement settings when writing network applications. The graph-based abstraction layer not only obtains the global network view but also presents it in a What You See Is What You Get (WYSIWYG) manner. The experiments in "Network Topology Verification", Graph Module Verification, "Data Persistence", and "Network Slice Verification" are based on this topology, as shown in Fig. 6A. We treat all network elements as nodes and the connections between them as edges in our model. The attributes of these nodes and edges are dynamically modified in real-time based on events occurring within the controller. This leads to a one-to-one correspondence between the attributes and the graph structure, allowing us to accurately represent the network topology.

Our graph-based abstraction layer is depicted using a drawing tool, as shown in Fig. 6B. This layer provides applications with the ability to access topological information, track changes in the topology, and load network statistics data efficiently. When new network elements arrive, we simply add a new node to the graph structure and establish the corresponding edges with its neighboring nodes. This process is triggered by specific events, such as "event.EventSwitchEnter" and "event.EventLinkAdd." Similarly, when network elements are removed, the abstraction layer updates the graph by deleting nodes and edges, responding to events like "event.EventSwitchLeave" and "event. EventLinkDelete." Furthermore, if there are changes in the attributes of network elements or links, the attribute values within the graph structure are updated accordingly. These updates are also event-driven, triggered by events such as "event.EventPortAdd" and "event.EventPortDelete".

The graph-based abstraction layer is done using the NetworkX module. During the experiment, we tested the proposed scheme based on two different random traffic patterns,

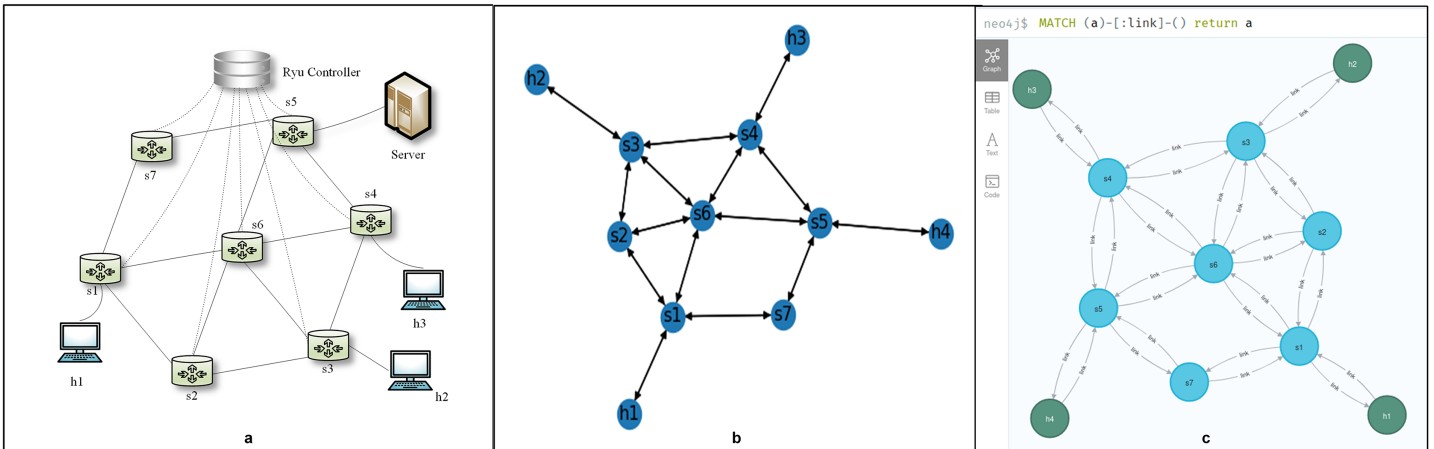

**Figure 6** The global network view. (A) Network structure. (B) Network topology of the graph-based abstraction layer. (C) Network element connection relationship in Neo4j.

elephant flow and mouse flow, in order to make it more realistic. We provided programs that can generate random flow of elephants and mice at any time. We also tested the performance of linear, mesh, and data-centre topologies (Fattree topologies). We successfully demonstrated the network topology structure through experiments, which provides information needed by SDN applications, such as topology, network statistics, and real-time topology updates. The graph-based abstraction indeed reduces the extra understanding and relationship conversion costs, which allows a network manager to write SDN applications without requiring mastery of low-level details.

## Graph module verification

NetworkX and Neo4j's ALGO extension both offer a variety of well-performing built-in graph algorithms. To demonstrate the use of these graph modules, we implemented shortest path forwarding based on link latency by leveraging the built-in weight-based shortest path functions. We obtained real-time latency for each link through delay detection and updated this latency as the edge weights in the graph abstraction layer. The graphical abstraction layer provides a global view, where we could use it to query the shortest path between two nodes by invoking built-in functions from the graph module. Once the path was determined, we queried the input and output ports for each hop along the path (port information is an attribute of the switch nodes). With this port information, we know how to deploy flow tables to guide packet transmission (refer to Fig. 5 for flow setup methods). In this way, shortest path forwarding based on link latency is achieved. As shown in Fig. 6A, there are multiple shortest reachable paths from node h1 to h2, such as h1 → s1→ s2 → s3 → h2 and h1 → s1 → s6 → s3 → h2. After the call graph module completes its calculations, the path h1 → s1 → s6 → s3 → h2, which had the shortest delay, was selected.

Based on other algorithms in the graph module, we could also perform a K-shortest path based on traffic flow; multi-path can be used for dynamic load balancing and can also

be used as a backup path for fault recovery. A network loop detection feature is available in the graph module to provide early notice of network loops. When a loop is detected, the minimal spanning tree of the graph can be used to choose a transmission path. We could also perform maximum flow detection, domain affiliation detection, and node importance judgment (PageRank algorithm).

Through experiments, we found that the graph-based abstraction layer has the following advantages when interacting with third-party modules:

- **Simplicity:** All data and attribute queries and modifications in the graph-based abstraction layer are intuitive and directly map network devices (Digital Twins), as shown in Code 2 (Eq. (2)).
- **Flexibility:** The graph-based abstraction layer is fully compatible with the Ryu architecture and third-party modules, as illustrated in Fig. 1, making it seem like an integral part of the system. This ensures great flexibility, allowing us to directly invoke built-in functions from the third-party module NetworkX to calculate the shortest path:

```
path = nx.shortest_path(self.Graph, source, target, weight = delay)
```

- **Scalability:** The graph itself is distributed. A node can be added to the graph, a graph can be added to another graph, and it can be infinitely extended. Similarly, deleting and updating sub-graphs is very convenient. Distributed processing can also be performed simultaneously. This shows that the graph-based abstraction layer is scalable, as shown in Code 3 (Eq. (3)).

## Data persistence

The graph-based abstraction layer manages the real-time state of the network, while data persistence in Neo4j accounts for its historical state. We chose the Neo4j graph database over a traditional relational database because it offers enhanced capabilities for storage, computation, and rapid relationship queries, which significantly enrich the abstraction layer. This choice allowed us to efficiently store and manage the corresponding network elements and their status within Neo4j as the graph-based abstraction layer is established. As illustrated in Fig. 6C, Neo4j effectively represents the relationships among network elements. The primary purpose of maintaining historical data is to enable network queries based on past time snapshots or intervals, facilitating an understanding of how network elements and status information have evolved over time. For instance, in scenarios such as network troubleshooting and service quality management, querying these time snapshots allows us to analyze the paths associated with network faults and observe their evolution.

In an SDN environment, we ran data persistence as a network application in the controller, which is essentially the same as other applications, such as access control and load balancing. It is a very special and important application. We use this method for data persistence, with time nodes as the initial nodes. The attributes to be recorded for each time node include timestamp, date, storage frequency, and note. "Time-to" and "time-from" relationships are used as edges between time nodes. "backup" is used as an edge between time nodes and network topology nodes. For how to import graphs from

NetworkX to Neo4j or, in other words, how to use Neo4j like NetworkX, refer to this link: https://github.com/jbaktir/networkx-neo4j.

Graph abstraction and graph databases are used to store data from the network. We can observe, predict, and learn using virtual networks in the past, present, and future. Then, interaction between the physical and virtual network, and the scenarios related to the network will be verified. This is an area that requires further research in the future.

## Network slice verification

In the architecture of this article, the graph abstraction was virtualized, and elements in the network can belong to a complex hierarchical classification. Here, we used access control to verify the feasibility of network slicing. Access control is also a commonly used controller application that determines which device is allowed to access and which is not. When implementing access control, the links that control the device's access also control the access request, and we used different flow table rules to restrict access from specific hosts or network interfaces.

In the design of the access control module, we stored the access control list (ACL) in the Neo4j. The ACL can be viewed as a whole network slice, with each row of the ACL being a node in the slice plane. The content of the node is stored as key-value pairs. The design quantifies access control into three steps: identifying a match, obtaining permission, and distributing the flow table.

1. **Identify match:** The program obtains host information from the abstract layer. Based on the source IP, destination IP, destination PORT, and other information, the program performs a record query in Neo4j to check if access permission exists. The list of permissions is shown in Table 1.

2. **Access control:** After the system program and the database have been matched up, it can be concluded whether the device has permission to access the resources it wants to access, and the permissions include Yes, No, and Exception (return exception information). If the permission is granted, a flow table is output to the SDN switch to allow the requested data from this device to be forwarded; if the permission is not granted, a flow table is output to the SDN switch to disallow (drop) the forwarding of the requested data for this device; if the permission is abnormal, the flow table is output to the SDN switch, and no processing is performed. The data distributed are shown in Fig. 7.

3. **Distribute flow table:** After the module is running, it will actively distribute the corresponding flow table rules according to the access permissions in the database. When the access permissions change in the database records, the flow table rules will be updated again. The system assigns a higher priority to ensure new flow table entries supersede previous ones within the validity period. When the flow table items expire, they are automatically removed. The terminal must re-acquire authorization to reaccess the resource server. We take advantage of the global nature of the topology to directly distribute the flow tables that need to be updated.

Due to the fact that access control involves frequent queries, we stored the ACL in Neo4j and verified the data persistence during network slice verification. The essence of access control is to implement the function of a packet-filtering firewall. With the flexibility of the

**Table 1 Permission record form.**

| no | src_ip | dst_ip | dst_port | acc_auth |
|----|--------|--------|----------|----------|
| 1 | 10.0.0.1 | 10.0.0.4 | 8001 | NO |
| 2 | 10.0.0.1 | 10.0.0.4 | 8002 | YES |
| 3 | 10.0.0.1 | 10.0.0.4 | 8003 | UNKNOWN |
| … | … | … | … | … |

```
1   data = {
2       "dpid": dpid,
3       "cookie": 0,
4       "cookie_mask": 1,
5       "table_id": 0,
6       "idle_timeout": 300,
7       "hard_timeout": 600,
8       "priority": 101,
9       "flags": 1,
10      "match": {
11          "dl_type": 0x0800,
12          "nw_proto": 6,
13          "tcp_dst": dst_port,
14          "nw_src": src_ip,
15          "nw_dst": dst_ip,
16      },
17      "actions": [
18          {
19              "type": "OUTPUT",
20              "port": 3
21          }
22      ]
23  }
```

**Figure 7 Distribute flow table data.**

SDN and the efficiency of the graph database queries, the final access control mechanism also has flexibility and high efficiency.

Although we have easily implemented the function of controlling access through network function virtualization, when various network functions are intertwined, network demand becomes more complex and dynamic. Further research is required for the functional verification of network slicing, which will be challenging. However, no matter which method is used, the way of managing network applications will change.

## Network latency discussion

### Experiment 1: Shortest path performance for various topology types in the graph-based abstraction layer.

The reason for the existence of the network is for traffic forwarding, and the main objective of traffic forwarding is to carry out the management of traffic forwarding paths.

We experiment on the selection of the shortest path in different topological networks to verify the availability of the graph-based abstraction layer. In order to better compare the experiment, we carefully selected three topologies that are very classic and can meet our special comparison requirements: linear, mesh, and fat-tree topology. We assume the following:

1) All topologies $K$ = 2, 8, 14, 20, …, 512 with a step size of 6. In linear and mesh topologies, $K$ denotes the number of switches. In the fat-tree topology, $K$ refers to the number of core layer switches. Given that the fat-tree topology consists of three layers of switches, we selected even numbers to meet the structural requirements of the topology.

2) In all topologies, we seek the shortest path from the first node to the last node. For example, for linear topology, when $K$ = 512, we seek the shortest path between node h1 to h512.

3) According to the implementation method of the graph abstraction layer, the edges of the topology must be bidirectional (have direction).

Linear topology is the simplest topology that can be complex; when $K$ = 512, there are 1,024 nodes and 2,046 edges, and querying the shortest path took 922.4414 µs, indicating a high query efficiency. According to the results in Fig. 8. the linear topology shows that the number of network hops increases synchronously as $K$ increases. Its shortest path delay is also steadily increasing, indicating that there is a positive correlation between the delay of the shortest path and the number of hops.

The mesh topology is a complex topological structure where each switch node is connected to all other switches, meaning that each switch can be directly connected. When $K$ = 512, it has the same number of 1,024 nodes as linear topology but has 262,656 edges. From the results in Fig. 9, it can be seen that although each node has 1,023 paths to reach the destination, the number of hops in a mesh topology network consistently remains at 1 regardless of how K increases. The average computation time is 151.6231 µs, indicating that the shortest path can always be found quickly. Its shortest path delay has a slight but insignificant increase as the network scale increases. The results indicate that the number of edges in the topology has a negligible impact on the shortest path latency, and the experimental results support our intuitive understanding.

Fat-tree topology has extensive applications in high-performance networks, data centers, enterprise networks, and campus networks. We need its complexity, as the network parameters increase exponentially with the growth of K, making it more complex than real-world network environments. When $K$ = 512, there are a total of 4,608 nodes and 1,056,768 edges. According to the results in Fig. 8, the fat tree topology has a fixed network hop count of 6 as K increases, and querying the shortest path took 167.1951 µs. Its shortest path delay increases slightly with the increase of the network scale, but not significantly. This suggests that the increase in the number of nodes and edges does not significantly impact the delay of the shortest path when the number of hops is fixed. The shortest delay of the 1-hop mesh network and the 6-hop fat tree network is almost the same, indicating that the impact of network hops on the delay is negligible.

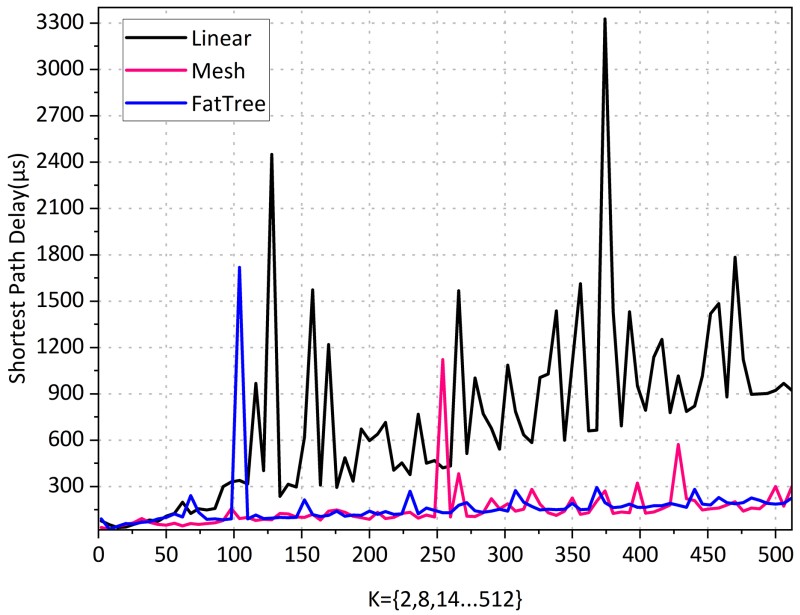

**Figure 8 Shortest path performance for various topology types in the graph-based abstraction layer.**

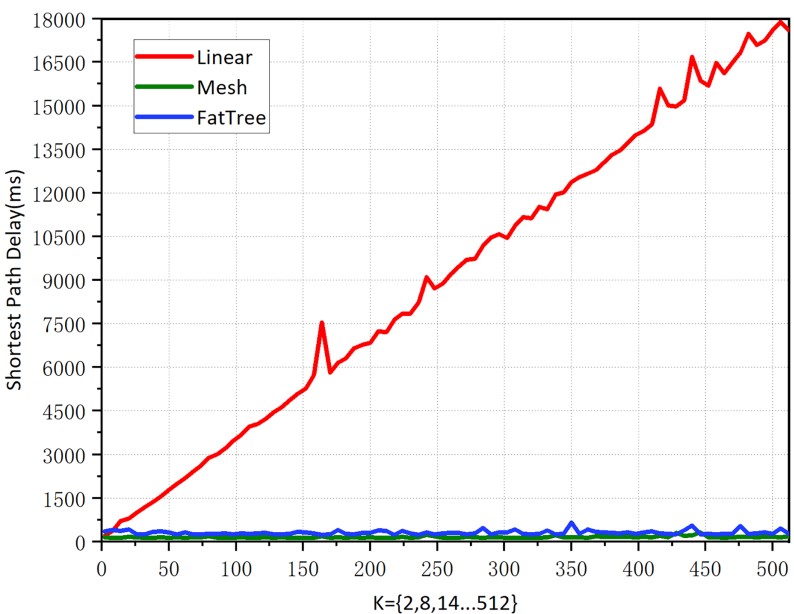

**Figure 9 Shortest path performance for various topology types in Neo4j.**

The graph-based abstraction layer has advantages in relationship lookup for professional details. Traditional relational database queries become increasingly difficult beyond the third level, and the difficulty coefficient starts to increase geometrically. Through program development, we have empirically examined these hypotheses, demonstrating the advantage of graph-based abstract approaches in the task of

relationships lookup. This advantage becomes more pronounced when dealing with complex and highly connected data. They can directly store and traverse relationships between nodes, avoiding the multiple JOIN operations in traditional relational databases, thereby potentially increasing efficiency.

**Experiment 2: Shortest path performance for various topology types in Neo4j.**

The abstract representation of graphs not only facilitates network understanding but also simplifies the formation of logical business combinations. However, this raises important questions: with the current trend towards integrating storage and computing in graph databases, is the graph abstraction layer becoming redundant? Given that all systems require data persistence, could a graph database itself serve as a replacement for the graph-based abstraction layer? In other words, why not incorporate the concept of a graph-based abstraction layer directly within the graph database? To address these questions, we conducted experiments to verify the necessity of maintaining a separate graph abstraction layer.

According to the statistics in Fig. 9, the linear topology shows a concurrent increase in the number of network hops as $K$ increases. The shortest path latency also increases proportionally. This indicates that the number of hops completely determines the latency, and when $K$ = 512, the shortest path query time is 17,603.67 ms.

Mesh topology increases as $K$ increases. Its shortest path is unchanged with the increase in network scale. This shows that the number of edges in the topology does not affect the shortest path's latency. When $K$ = 512, the shortest path query time is 185.28 ms.

Fat tree topology as $K$ increases. Its shortest path barely changes with the increase in network size. This shows that the increase in the number of nodes and edges, with a fixed number of hops, has no effect on the latency of the shortest path. When $K$ = 512, the shortest path query time is 310.44 ms.

After comparing Figs. 8 and 9, we found that the latency of the graph-based abstraction layer is significantly smaller than that of Neo4j. We averaged the shortest path computation time for different topologies ranging from $K$ = 1 to 512 in our experiments and found that the average computation time for Neo4j (3,183,508.6 ms) is 9,121.81 times greater than that of the graph-based abstraction layer (348.9996 µs). The graph-based abstraction layer is computed in memory, which is naturally faster than querying data stored in a database. The abstraction layer acts as a cache between the CPU and the database. In practical network management, we need to frequently access the current state in memory, which also highlights the necessity of the graph abstraction layer. Historical data is stored in the graph database Neo4j, and it is not appropriate to completely replace the graph-based abstraction layer with the graph database.

## Overload analysis

In this experiment, we measured the overhead of graph-based abstraction layer, mainly referring to CPU utilization and memory usage. We confirmed in our experiment that the graph-based abstraction layer indeed consumes minimal system resources. For the CPU utilization of the graph abstraction layer, we tested the overhead with and without the

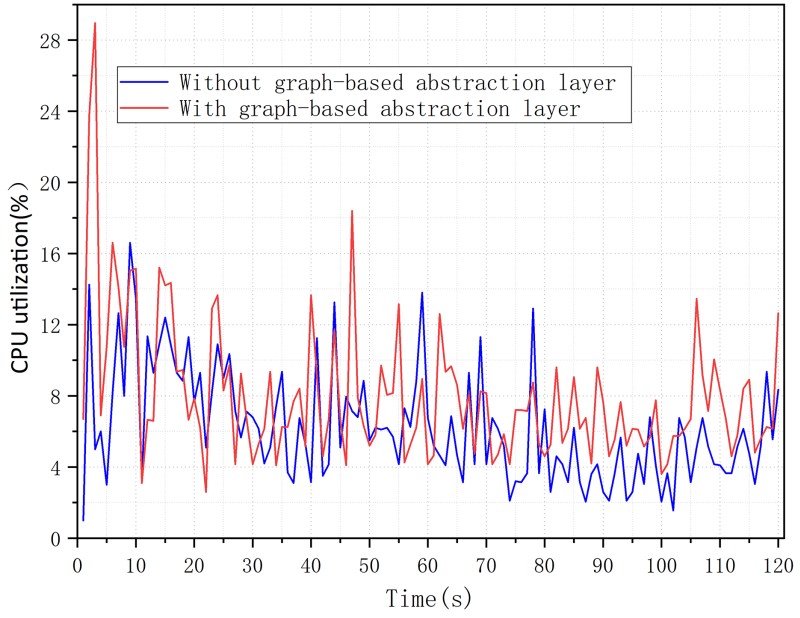

**Figure 10  CPU utilization with a graph-based abstraction layer.**

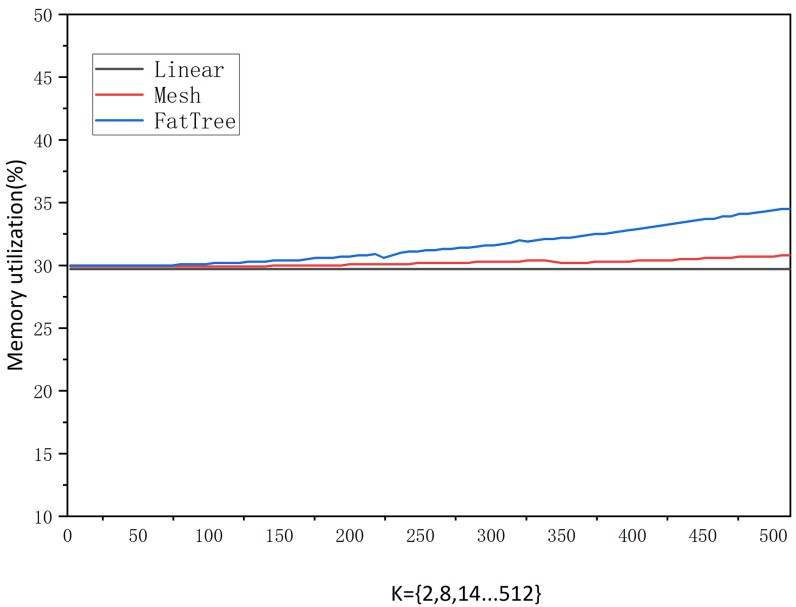

**Figure 11  Memory occupancy.**

graph abstraction layer. Figure 10 shows that the average CPU utilization increased from 6.22% to 7.99% over 120 s, without introducing significant CPU utilization. It was also found that there was a spike in utilization only at the initial loading phase, while at other times the increase in utilization might be due to computational overhead caused by updates to nodes and attributes.

We also measured memory usage, and Fig. 11 shows that the memory usage grows very gradually under different traffic sizes and network structures. From $K = 2$ to 512, it can be observed that the linear topology shows no growth, the mesh topology grows by 0.9%, and the fat-tree topology grows by 4.5%. This indicates that the size and variation of traffic in the network do not affect memory usage. On the other hand, we can also see that memory growth is only related to the number of network nodes and edges. When $K$ is fixed, memory is allocated directly based on the number of nodes and edges. Subsequent changes in traffic only alter edge attribute values without affecting memory. However, the addition or removal of nodes, edges, and attributes may slightly influence memory usage.

## CONCLUSION

Upon investigation, there have been few studies on the SDN graph-based abstraction layer in the existing literature. We propose a virtual architecture based on the characteristics of SDN. Graph abstraction makes good use of programmable and virtualized features. It can be well summarized and used in more complex scenarios, which is a high-level abstraction that conforms to human intuition. Our graph-based abstraction layer implementation provides network topology, network resources, and network information flows (delay, loss, throughput, bandwidth, current rate) to the SDN control plane and application plane. This intuitive, natural, real-time representation, with good readability and reusability, is very conducive to universal development.

Existing SDN controllers lack a standard data model. Interoperability between heterogeneous SDN controllers and applications can only be achieved through the standard northbound and east-west interfaces. The graph-based abstraction layer allows applications to only update the graph itself without having to update a large number of tables. The complexity and latency can be significantly reduced when the query data of network applications has a correlation and more jumps.

We have demonstrated the feasibility of graphical abstraction in topological display, dynamic route, access control, and data persistence. There is no need to change the controller's source code.

The graph-based abstraction layer can extend the virtualization of SDN and lay the foundation for future work. Data persistence can be observed, predicted, and learned using virtual networks in the past, present, and future. An analysis will be conducted to validate the network possibilities. This work can be easily extended to other controllers or adapted to other types of applications, leaving such extensions for future work. For others to try, the core graph-based abstraction layer source code is open source (https://github.com/Jyfeng2021/SDN-graph-based-abstraction-layer.git) to facilitate various SDNAPP extensions based on this work.

## ACKNOWLEDGEMENTS

The authors are grateful for the valuable comments and suggestions of the reviewers.

### Funding

This research was funded by the Foundation of Xinjiang College of Science and Technology (2024-GGYX03). The funders had no role in study design, data collection and analysis, decision to publish, or preparation of the manuscript.

### Grant Disclosures

The following grant information was disclosed by the authors:
Foundation of Xinjiang College of Science and Technology: 2024-GGYX03.

### Competing Interests

The authors declare that they have no competing interests.

### Author Contributions

- Yufeng Jia conceived and designed the experiments, performed the experiments, analyzed the data, performed the computation work, prepared figures and/or tables, and approved the final draft.
- Jiadong Ren conceived and designed the experiments, performed the experiments, authored or reviewed drafts of the article, and approved the final draft.
- Xianshan Li analyzed the data, authored or reviewed drafts of the article, and approved the final draft.
- Haitao He analyzed the data, authored or reviewed drafts of the article, and approved the final draft.
- Pengwei Zhang performed the computation work, prepared figures and/or tables, and approved the final draft.
- Rong Li analyzed the data, prepared figures and/or tables, and approved the final draft.

### Data Availability

The code is available in the Supplemental Files, GitHub, and Zenodo:
- https://github.com/Jyfeng2021/SDN-graph-based-abstraction-layer.git
- Jyfeng2021. (2025). Jyfeng2021/SDN-graph-based-abstraction-layer: graph-based abstraction layer v1.0 (1.0). Zenodo. https://doi.org/10.5281/zenodo.14683957.

### Supplemental Information

Supplemental information for this article can be found online at http://dx.doi.org/10.7717/peerj-cs.2674#supplemental-information.

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
