# Peer review of "Construction of a user-friendly software-defined networking management using a graph-based abstraction layer"

_PeerJ Computer Science, doi:10.7717/peerj-cs.2674_

## Round 0.1 · original submission · Major Revisions

Dear authors,

You are advised to critically respond to all comments point by point when preparing an updated version of the manuscript and while preparing for the rebuttal letter. Please address all comments/suggestions provided by reviewers, considering that these should be added to the new version of the manuscript.

Kind regards,

PCoelho


**Language Note:** The review process has identified that the English language must be improved. PeerJ can provide language editing services - please contact us at [email protected] for pricing (be sure to provide your manuscript number and title). Alternatively, you should make your own arrangements to improve the language quality and provide details in your response letter. – PeerJ Staff

·

Basic reporting

This paper presents an architecture to represent the network as a graph. The purpose of this architecture is to implement an abstraction of the SDN controller at the application plane. The authors abstract all network elements with a graph model, with the attributes of the elements as the attributes of the
graph.

Experimental design

I like the experiments of the paper.

Validity of the findings

The authors put a section for Results and discussion and the discuss the work very well.

Additional comments

The only comment I have is that the authors put a reference in Conclusion part, please remove it.

Reviewer 2 ·

Basic reporting

The paper presents a novel approach to simplifying the management of software-defined networking (SDN) through a graph-based abstraction layer, offering an intuitive and efficient method for representing network elements and relationships. This approach addresses significant challenges in the lack of standardized northbound interfaces and the complexity of SDN management. The authors validate their solution through comprehensive experimentation, demonstrating its feasibility in network topology verification, delay-based shortest path forwarding, and data persistence. The theoretical foundation is strong, with clear connections to established SDN frameworks like Ryu, Neo4j, and Mininet, adding credibility and practical relevance to the proposed method. Furthermore, the clear problem definition and innovative solution make this paper a valuable contribution to the field of SDN.

Experimental design

The paper suffers from many weaknesses and I propose some areas for improvement :

Methodology Details: While the paper presents a promising approach, more detailed questions arise regarding the implementation of the graph-based abstraction. How exactly are physical network elements transformed into graph representations? Could the process of assigning attributes like delay, bandwidth, and throughput to edges be better explained to ensure replicability? A deeper, step-by-step breakdown would improve clarity.

Experimentation Metrics: The paper provides some performance metrics, but how does the proposed solution perform under a broader range of conditions, such as higher traffic volumes or larger network topologies? Are there additional metrics, like memory usage or network throughput, that could offer a more comprehensive assessment of the system's efficiency and scalability?

Data Persistence: The paper discusses data persistence using Neo4j, but how does this solution hold up under stress, such as in high traffic conditions? Could potential bottlenecks or retrieval delays in accessing historical network data from Neo4j be explored more critically? What are the performance trade-offs between real-time memory-based calculations and database storage for large-scale networks?

Broader Applicability and Future Work: While the graph-based abstraction layer works well within the scope of the presented experiments, how easily could this framework be adapted to other SDN controllers or integrated with new technologies like AI-driven automation? Could future research explore the scalability and adaptability of this solution to more varied network environments and use cases?

Validity of the findings

Validation Across Different Scenarios: The current validation is limited to a few network topologies and traffic types. How would the proposed abstraction layer perform in more complex, real-world network environments? Could the authors explore how it handles diverse traffic patterns, such as bursty or unpredictable traffic loads, and different real-world topologies?

·

Basic reporting

The authors propose an abstraction layer of the control plane towards the SDN northbound interface in order to represent the underline network as a graph. The introduction of the paper is very clear and the work is well-motivated: the literature still lacks a suitable northbound interface abstraction and standard.

On the contrary, the rest of the paper is very unclear and the writing becomes disjointed and incomprehensible in most parts. Some examples where the language must be improved
include lines: 208-217, 234-240, 305-308, 339-346, 336-372, 422-428, 447-452, 468-471, 479-485.
This aspect is the major shortcoming of the article.

As a consequence, the tests performed and the evaluation of the results are not clear. In particular, only the "Network slice verification" section is understandable, showing a clever application of the proposed solution.

While I thank the authors for providing the source code in a GitHub repository, it is very hard to analyze and run the code considering that the instructions are in Chinese. This could have helped in understanding the author's work, even if the paper is unclear. I suggest the authors update the code in the GitHub repository by adding instructions in English.

Finally, the figures' colors are confusing and with too little contrast. Sometimes, what is represented in the figure is not explained in the test nor in the figure's caption. Take Figure 5 as an example.

Experimental design

The research questions are well-defined, relevant, and meaningful. On the contrary, the methodology, design, and solution are unclear.

The experimental environment and the technologies employed are well-described.

The tests, results, and discussion must be improved.
- First of all, the authors should clarify Section 4.2. What are the attributes and events referred to in line 310? How frequent is the addition and removal of network elements?
- In the overload analysis, why only CPU utilization is considered? The authors should also describe if this metric changes when the number of nodes in the topology increases.
- In experiments 1 and 2, which is the ground truth?
Still, Section 4.5 is a good contribution and presents an application of the proposed abstraction layer that meets the analyzed research questions.

Validity of the findings

By enhancing the comprehensibility of the paper, the validity of the findings can be better assessed. At the current state of the article, it is very difficult to assess them.
Additionally, the authors should highlight the ground truth of the performed experiment.

---

## Round 0.2 · Minor Revisions

Dear authors,

Thanks a lot for your efforts to improve the manuscript.

Nevertheless, some concerns are still remaining that need to be addressed.
Like before, you are advised to critically respond to the remaining comments point by point when preparing a new version of the manuscript and while preparing for the rebuttal letter.

Kind regards,
PCoelho

·

Basic reporting

The authors did a great job

Experimental design

The authors did a great job

Validity of the findings

The authors did a great job

Additional comments

The authors did a great job

·

Basic reporting

The authors have addressed my previous comments adequately and completely. The revisions have significantly improved the paper's clarity, particularly in the sections that were previously disjointed and difficult to follow. The language is now clearer, and the explanations are coherent, making the methodology and results much easier to understand.

I appreciate the efforts made to enhance the visual quality of the figures, which now effectively support the narrative of the manuscript. Additionally, the inclusion of English instructions in the GitHub repository has greatly improved the reproducibility and accessibility of the work. However, I suggest that the authors translate all the code comments into English to further enhance usability and clarity.

With these improvements, I was able to assess the paper more accurately and fully understand the proposed contribution. While I recognize that this is a very minor and limited contribution to the state of the art, I believe it aligns with the journal's criteria and may still be considered for publication by the editor.

Experimental design

no comment

Validity of the findings

The major shortcomings that the article still suffers are as follows:
1) The article still lacks evaluation with multiple controllers (only Ryu is considered), which would be necessary to strengthen its claims of generality and applicability.
2) Additionally, the evaluation is weak in some parts: the authors define the size range of random traffic (mouse and elephant), but it seems they never actually use it in the experiments in Sections 4.2, 4.3, 4.4, 4.5, and 4.6. This omission reduces the robustness and relevance of the experimental results.

Additional comments

Some minor issues need attention:
1) Some parts still require improvements in English, and there are punctuation errors (e.g., full stops without a subsequent space).
2) There is a certain degree of formula occlusion in Section 3.2 at lines 218 and 222, which hinders readability.
3) The authors sometimes refer to "Code," but there are only figures in the paper (e.g., line 386), which is confusing for the reader.

These issues should be addressed to ensure the paper meets the necessary standards before publication.

---

## Round 0.3 · accepted · Accept

Dear authors, we are pleased to verify that you meet the reviewer's valuable feedback to improve your research.

Thank you for considering PeerJ Computer Science and submitting your work.

Kind regards
PCoelho

·

Basic reporting

The authors have addressed my previous comments completely.

Experimental design

no comment

Validity of the findings

no comment

Additional comments

no comment